# Case Series of Disseminated Xanthogranulomatosis in Red-crowned Parakeets (*Cyanoramphus novaezelandiae*) with Detection of Psittacine Adenovirus 2 (PsAdV-2)

**DOI:** 10.3390/ani12182316

**Published:** 2022-09-06

**Authors:** Cornelia Konicek, Kristin Heenemann, Kerstin Cramer, Thomas W. Vahlenkamp, Volker Schmidt

**Affiliations:** 1Service for Birds and Reptiles, Clinic for Small Animal Internal Medicine, University of Veterinary Medicine Vienna, Veterinärplatz 1, 1210 Wien, Austria; 2Institute of Virology, Faculty of Veterinary Medicine, University of Leipzig, An den Tierkliniken 29, 04103 Leipzig, Germany; 3Clinic for Birds and Reptiles, Faculty of Veterinary Medicine, University of Leipzig, An den Tierkliniken 17, 04103 Leipzig, Germany

**Keywords:** psittaciformes, siadenovirus, PsAdV-2, lipometabolic disorder, xanthoma, parakeet, adenovirus

## Abstract

**Simple Summary:**

Lipometabolic disorders, such as xanthogranulomatosis, are common diseases in avian medicine. Various manifestations of lipometabolic disorders and risk factors for acquiring lipometabolic diseases have been described in the past years. Xanthogranulomas are usually limited to the skin and supposed to be associated with traumatic or inflammatory injuries in that area. Disseminated xanthogranulomatosis, appearing simultaneously in several internal organs, has been recently described in psittacine birds, the cause of the diseases was not known. Here, we describe a case series of disseminated xanthogranulomatosis in another psittacine species, the Red-crowned Parakeet (*Cyanoramphus novaezelandiae*) and a possible association with a concurrent psittacine adenovirus 2 (PsAdV-2) infection. Viral infections that trigger lipometabolic diseases have been described in human medicine in some species of small animals and in chickens. PsAdV-2- infections are widely distributed in avian species. A possible association between PsAdV-2- infections and lipometabolic diseases in the Red-crowned Parakeet should be considered. Individual birds and flocks with both or either of these diseases should be carefully examined and monitored.

**Abstract:**

Xanthogranulomatosis is a common dermatological disease in birds. This form of inflammation, possibly associated with lipometabolic disorders, can also be seen in visceral organs, which as yet has only rarely been described in avian medicine. In general, diseases related to impaired lipid metabolism are frequently reported in avian medicine, with hepatic steatosis and atherosclerosis being the most common. In human medicine, infectious agents—especially some strains of adenovirus—were implicated in contributing to lipometabolic disorders; this has also been described for chicken. Here, a case series of six Red-crowned Parakeets (*Cyanoramphus novaezelandiae*) is presented, all cases being characterized by psittacine adenovirus 2 (PsAdV-2) infection with or without disseminated xanthogranulomatosis. The affected individuals were examined alive by clinical examination. Total body radiographs were taken of two birds, haematology and blood biochemistry results were achieved in one bird. The birds either died immediately after clinical presentation or within two days, two individuals were euthanized due to worsening of their clinical condition. All birds underwent a post-mortem examination. While four birds were finally diagnosed with disseminated xanthogranulomatosis, all six individuals had large eosinophilic intranuclear inclusion bodies in the epithelial cells of the collecting ducts of the kidney and tested positive for PsAdV-2. Further examinations are needed to clarify to what extent PsAdV-2 might elicit lipometabolic disease in birds, or psittacines in general, and, in particular, the Red-crowned Parakeet.

## 1. Introduction

The Red-crowned Parakeet (*Cyanoramphus novaezelandiae*) is a species endemic to New Zealand, currently listed as a species of least concern by the International Union for the Conservation of Nature, although numbers are decreasing (https://www.iucnredlist.org/, accessed on 28 November 2021). They are commonly kept as pet birds and frequently held in aviculture in Germany. Commonly reported diseases are mycobacteriosis [1,2], circovirus infection [3,4], haemosporidiosis [5,6], *Cryptosporidium avium* [7] and psittacine adenovirus 2 (PsAdV-2), which has subsequently been referred to as psittacine siadenovirus F [8]. Avian adenoviruses are currently classified in three different genera: atadenovirus, aviadenovirus and siadenovirus [9,10]. They have been associated with a broad range of lesions in various species (such as hepatitis, gastritis, enteritis, splenitis, nephritis, conjunctivitis, pneumonia), and linked to acute mortality events, identified as part of multifactorial diseases or described as incidental findings without correlating pathologies [11,12,13]. It has been suggested that adenovirus-related lesions in psittacines reflect subclinical infections in the face of immune suppression [12]. The apparent ability of avian adenovirus to cause a high prevalence of subclinically infected, actively shedding birds (and to infect a range of species) implies that dissemination of this virus is likely to be much greater than previously recognized [14].

So far, PsAdV-2 infections have been reported in nine psittacine species from three regions, Australia, New Zealand and the Indian subcontinent [8,12,14,15,16,17]. Just recently, PsAdV-2 infection has been found in an African Grey Parrot (*Psittacus erithacus*) with accompanying clinical signs and pathological findings [18]. While avian siadenovirus are supposed to have a wide host range [12] and are reportedly widely circulating in avicultural collections [14], the full spectrum of species susceptible is unknown as yet, which merits further investigation [14].

Human adenovirus 36 (Ad-36) (mastadenovirus) is linked to lipometabolic disorders in humans and chickens and experimentally in mice and marmosets [19,20,21], while Ad-5 increases adiposity in rats [22]. The E4orf1 gene of Ad36 was identified as necessary and sufficient to induce adipogenesis in cells [23]. However, a chicken model that accumulates fat upon infection with an avian adenovirus SMAM-1 virus was described already some years earlier from India [24]. Lipometabolic disorders in psittacine birds, such as atherosclerosis, hepatolipidosis, lipoma, steatitis or xanthomas, are seen in avian practice on a regular basis [25]. Xanthogranulomatosis is most often restricted to the skin and regularly found in Cockatiels (*Nymphicus hollandicus*) and Budgerigars (*Melopsittacus undulatus*). These lesions are non-neoplastic, but focally infiltrative and appear as variably sized, yellow to orange nodules or plaques, formed from lipid deposits [26]. Xanthogranulomas tend to develop in areas where physical trauma, local pressure, bleeding or inflammation have occurred or are still occurring, but rarely in internal organs, and has even more rarely been identified within the bone marrow [27]. Just recently, a form of disseminated xanthogranulomatosis has been described in Eclectus Parrots (*Eclectus roratus*) and budgerigars [28,29]. In these case collections, no potentially causative inflammatory agent was identified, nor was there any other possible cause other than inadequate nutrition and husbandry.

We examined a case series of disseminated xanthogranulomatosis in privately kept Red-crowned Parakeets, a rare disease, which has not yet been reported in this species. We suspect a possible connection with a simultaneously detected PsAdV-2 infection.

## 2. Materials and Methods

Samples used in this study were collected as part of disease investigations in all six birds, since a diagnostic necropsy was requested in each case by the respective bird owners. The Animal Ethics Committee at the University of Leipzig was informed that findings from the diagnostic material were to be used in a publication and a formal waiver of ethics approval was granted.

Case data of six Red-crowned Parakeets kept in four different households were included in this study, each collected retrospectively over a period of three years (2018–2021). All birds were presented alive at the Clinic for Birds and Reptiles, University of Leipzig, Germany, because of various clinical signs. Inclusion criteria for the study were, beside the species, diagnosis of disseminated xanthogranulomatosis or PsAdV-2 infection.

All individuals were privately kept, the majority in indoor aviaries with daily free flight in the household. Only two birds kept in one flock had access to an outdoor aviary. All of the birds were young adults from ten months to a maximum of four years of age, four were males and two were females. Nutrition was similar in all cases and consisted of commercially available seed-based diet, vegetables and fruits. All birds underwent a physical examination, total body radiographs were taken of two of the birds, haematology and blood chemistry was run in one bird. Three birds died within a few hours after hospitalisation, one after two days, and two were euthanized due to worsening of their clinical condition. Where birds were kept in the same household, the two individuals died within two months (case nos. 3 and 4) and five months (case nos. 5 and 6), respectively.

Full gross necropsy, including cytological, histopathological, parasitological, bacteriological and mycological examination as described elsewhere, was performed in all cases [30]. Impression smears from spleen, liver, lung, crop/oesophagus, proventriculus, small and large intestine were stained with RAL Diff-QuikTM (RAL Diagnostics, Martillac, France) as well as with a Ziehl–Neelsen stain and examined microscopically at 1000x magnification. Sections of skin, conjunctiva, bone marrow, visceral organs and brain were fixed in 4.5% neutral-buffered formalin for at least 24 h. Formalin-fixed samples were dehydrated, routinely embedded in paraffin wax and sectioned at 4 µm. All sections were stained with haematoxylin and eosin, Congo red and Von Kossa stains as well as periodic acid–Schiff (PAS) stain. Ziehl–Neelsen and Fite-Faraco stain were applied to identify acid-fast bacteria. In order to identify fungal pathogens, Gomori’s methenamine silver stain was used.

Collected organ samples were stored at −20 °C for further molecular diagnostic of viral pathogens. In each case, DNA was isolated from 10 mg splenic, 15 mg liver and 15 mg kidney tissue and, when present, 15 mg of the xanthogranulomatous masses by use of DNeasy Blood & Tissue Kit (Qiagen, Hilden, Germany). In order to test for adenovirus, polyomavirus and circovirus, family-specific consensus-nested PCRs were performed as previously described [31,32,33]. Positive PCR products were purified using GeneJET PCR Purification Kit (Thermo, ScientificTM, Schwerte, Germany) and subsequently sent to Microsynth Seqlab (Göttingen, Germany) for Sanger sequencing. Nucleotide sequences were analyzed and edited using GENtle program (Magnus Manske, University of Cologne, Germany). Subsequently, obtained data were compared under use of the National Database for Biotechnological Information (NCBI; https://www.ncbi.nlm.nih.gov/, accessed on 21 May 2022).

## 3. Results

### 3.1. Clinical Findings

Owners of the six Red-crowned Parakeets reported various, mostly unspecific clinical signs, including apathy, reduced appetite or anorexia. Two birds were presented because of soft tissue masses on the knees (case no. 1) (Figure 1) and on the neck (case no. 2), respectively. A third bird had nasal and conjunctival discharge. Concerning their nutritional status, two birds were emaciated, and the remaining four had a reduced body condition. Body weights ranged from 49 g to a maximum of 79 g. Overgrown beak and claws were seen in three birds and feather loss in one bird (Table 1).

One Red-crowned Parakeet (case no. 1) with visceral xanthogranulomatosis had a marked leukocytosis (31.3 × 10 E9/L) with heterophilia (18 × 10 E9/L) and monocytosis (5.8 × 10 E9/L). Cholesterol (7.88 mmol/L) and low-density lipoprotein (LDL) (3.29 mmol/L) were increased and aspartate aminotransferase (AST) (16.4 U/L) decreased. Other parameters of the haematological examination and clinical chemistry were within the expected ranges.

Radiographic findings in case no. 1 were soft tissue masses in the region of both knees (distinctly more pronounced on the left side, stretching from mid-femur to mid-tibiotarsus), overlying the proximal right femur (stretching to the hip), as well as the right distal femur presented distended, with irregular decreased bone density and multiple osteolytic defects. Another area with increased radiodensity, of granulomatous-like appearance, could be seen in association with the caudal edge of the right lung (Figure 2). In case no. 4, widening of the cardiohepatic silhouette (esp. of the caudal part, loss of hourglass shape) was visible.

### 3.2. Gross Necropsy Findings

Visceral white, firm nodules of up to 3 cm in diameter were found in various tissues and organs of four birds (cases nos. 1, 2, 4, 6). Multifocal fascial nodules between 0.1 cm up to 1.6 cm on both knees [Figure 1a], cranial to the right *Os ileum* and at both acetabuli as well as in the lung, apical in the pericardium and in the liver were seen in case no. 1 [Figure 1b]. Multiple nodules measuring 1.0 to 3.0 cm in diameter were seen in the fascia of the neck, between muscles (*M. pectoralis* und *M. supracoracoideus*), the serosa of the thorax, the mesenterium, as well as in the right lung and the liver in case no. 2. Multifocal white nodules with a diameter of 0.2 cm were located in the mesenterium, pericardium, as well as in liver and spleen in case no. 4, and in the lung and air sacs in case no. 6 (Table 1). Additional findings were cloudy nasal and conjunctival exudate with bilaterally thickened eyelids in one bird (case no. 6), as well as an infestation with *Procnemidocoptes janssensi* resulting in feather loss in one bird (case no. 1). Nodular alterations were absent in two birds, one of them showing hepatosplenomegaly, a friable lung (unilaterally) and two indented black foci in the proventriculus measuring up to 0.2 cm in diameter (case no. 3), the other, however, presented no gross pathological findings (case no. 5) (Table 1).

The nodules from birds with case nos. 1, 2, 4, 6 consisted cytologically of foamy cells, macrophages, heterophiles, lymphocytes and an amphophilic homogenous material. Impression smears in case no. 3 revealed reactive lymphocytes in spleen and liver, as well as heterophiles, lymphocytes, histiocytes and fibrin in the lung. Aspergillus fumigatus was isolated in this case from the lung. Vacuoles in the liver impression smear were seen in case no. 5. Endoparasites or bacteria were not isolated, neither were acid-fast bacteria detected in any case.

### 3.3. Histopathological Findings

The nodules from birds with case nos. 1, 2, 4, 6 were characterized histopathologically by diffuse foamy and cholesterol-laden macrophages, multifocal fibrinous granulomas, central necrosis and fibrin, surrounded by multinucleated giant cells, histiocytes, lymphocytes and heterophils as well as various degrees of fibroplasia. Ziehl–Neelsen, Fite-Faraco, Von Kossa and Kongo red stains as well as PAS reaction were without additional findings, thus identifying these lesions as xanthogranulomatous inflammations [27,34] (Figure 3). In all six birds, large eosinophilic intranuclear inclusion bodies were documented in the epithelial cells of the collecting ducts of the kidney (Figure 4), regardless of the diagnosis of xanthogranulomas, which were present in four birds only. Additional findings in birds with xanthogranulomas (case nos. 1, 2, 4, 6) were splenic and renal haemosiderin in case no. 1, and subacute fibrinous conjunctivitis with isolation of *Pseudomonas aeruginosa* in case no. 6 (Table 1). The main histopathological findings in both birds without xanthogranulomas (case nos. 3, 5), in addition to viral nephritis, in case no. 3, comprised disseminated intravasal coagulation, necrotizing splenohepatitis with amphophilic intranuclear inclusion bodies inside reticular cells and hepatocytes, lymphocytic depletion of the lymph follicles of the spleen, acute catarrhal duodenitis, subacute ulcerative proventriculitis and subacute fibrinous fungal pneumonia, and in case no. 5, hepatic steatosis (Table 1).

### 3.4. Virological Results

All six Red-crowned Parakeets were tested positive for adenovirus-DNA, irrespective of the diagnosis disseminated xanthogranulomatosis. Sequencing of all obtained amplificants yielded identical results. All sequences were deposited in NCBI GenBank (NCBI accession no.: OM994429-OM994433). NCBI Blast analysis revealed the highest identity of 97.64% to PsAdV-2 strain WVL19065-01E (NCBI accession no.: MZ562791.1).

### 3.5. Phylogenetic Analysis

Phylogenetic analysis. Partial polymerase gene sequences (311 position, shorter sequences were excluded) were analyzed in comparison to sequences of 62 reference gene sequences from NCBI (Figure 5). The phylogenetic tree was constructed using MEGA X software [35], Maximum Likelihood method and Jukes–Cantor model [36]. The bootstrap consensus tree inferred from 1000 replicates [37] is taken to represent the evolutionary history of the taxa analyzed [38]. The unrooted trees have a site coverage cutoff of 95%. Branches corresponding to partitions reproduced in less than 50% bootstrap replicates are collapsed. The percentage of replicate trees in which the associated taxa clustered together in the bootstrap test (1000 replicates) are shown next to the branches [38]. Codon positions included were 1st + 2nd + 3rd + Noncoding.

## 4. Discussion

Reports on disseminated xanthogranulomatosis in birds are rare. Recently, a case series on Eclectus Parrots and Budgerigars was published [28]. In most cases of xanthomatosis in birds, which occurs frequently in psittacine as well as in gallinaceous birds [34], lesions are found locally limited to the skin. There are reports on tracheal xanthogranulomatosis and periosseus xanthogranulomatosis in a Red-tailed Hawk (*Buteo jamaicensis*) [37] and a fledgling of a Great Horned Owl (*Bubo virginianus*) [26], respectively, as well as a conjunctival xanthoma in a Blue-and-yellow Macaw (*Ara ararauna*) [39].

Here, we present four cases of disseminated xanthogranulomatosis in Red-crowned Parakeets, which has not been described before in this species. Clinical findings were rather unspecific: only two birds had additional subcutaneous xanthogranulomatous masses, suggestive of a granulomatous diseases complex. These findings are in line with the previously published cases of Eclectus Parrots and Budgerigars [28,29].

In humans, these inflammatory processes are seen in different anatomic locations, especially in the kidneys, gall bladder, genital tract, and the appendix [40,41]. There is no single specific aetiology for this form of inflammation. It is attributed to several causes in humans, such as persistent chronic infection, suppuration, necrosis, haemorrhage, obstruction, lipometabolic disorders, calculi or foreign bodies [42]. However, the majority of xanthogranulomatous pyelonephritis seem to be associated with bacterial infection [41]. In the present study, routine microbiological examinations were performed, but no corresponding bacterial infection could be identified.

In regard to the occurrence of cutaneous xanthogranulomas in avian medicine, various aetiological theories have been proposed, including high-lipid diets [34]. The exact composition of the seed mixture offered to the birds by the bird owners was not evaluated in this study. Given that seed-based diets are known to be unbalanced for psittacine birds and often high in fats [43], this nutritional imbalance may be a predisposing factor for acquiring xanthogranulomatous diseases, as stated before [28]. In contrast, recently published experimental studies in Monk Parakeet (*Myiopsitta monachus*) provide some evidence that in the genesis of atherosclerotic lesions, cholesterol concentration of the diet is more important than the overall fat content [44]. Lipometabolic disorders are suspected to be the main reason for the pathogenesis of disseminated coelomic xanthogranulomatosis, since concurrent atherosclerotic lesions were seen [28,29]. Atherosclerotic lesions were absent in all six Red-crowned Parakeets investigated here. While this species is not classified as highly susceptible for the development of atherosclerosis [45], it could well be that lipometabolic disorders in Red-crowned Parakeets are more prone to result in disseminated xanthogranulomatous inflammation than in acquiring atherosclerosis. To the authors’ knowledge, so far, no reports of atherosclerosis in Red-crowned Parakeets exist.

Reference intervals for blood biochemistry were not available for this species. Therefore, the Species360 database (https://www.species360.org/, accessed on 21 December 2021) was searched for a closely related species, namely, *Platycercus eximius*. Values recorded in this database, however, originate from only few animals, and not all of the values analyzed in the present study are included. Comparison of values to established ranges for other psittacine species [46] suggests that the cholesterol level might be slightly elevated in case number one. Concentrations of LDL were measured by enzymatic method, but this method is not validated in avian medicine. Reference values obtained when using an enzymatic method are available for the Common Myna (*Acridotheres tristis*) (1.11 mmol/L–2.64 mmol/L) [47]. There are also reference values for *Amazona* spp. (2.84 mmol/L–3.48 mmol/L) and *Psittacus* spp. (2.9 mmol/L–3.63 mmol/L) calculated by the Friedewald formula [48], but there is a known bias for the calculated values [49]. Considering these limitations and the fact that blood values were only available in one case, no definite conclusions can be drawn from these data. At least, there were marked leucocytosis with heterophilia, suggesting an inflammatory process.

Two birds with disseminated xanthogranulomatosis had abnormal but inconclusive radiographic findings, with only one bird showing distinct soft tissue masses. In the study from Hanson et al. (2020) [29], common imaging features suggestive of this disease process include coelomic effusion, multifocal heterogeneous, focally mineralized nodules/masses (predominantly coelomic in localization) and hepatomegaly. In conclusion, the authors stated that imaging features of xanthogranulomatous inflammation are nonspecific and thus difficult to recognize [29], which is in line with the present study’s results.

Considering the number of cases that were collected in quite a short period of time, a predisposition for xanthogranulomatous inflammation in the Red-crowned Parakeet may be hypothesized. Clinical diagnosis of disseminated xanthogranulomatosis is difficult to achieve. As clinicians are usually more familiar with other relevant and more common diseases in psittacines that present as disseminated visceral nodules, such as aspergillosis, mycobacteriosis and neoplastic diseases such as lymphoma, a thorough diagnostic work-up should be performed in each of these cases, including cytological, microbiological, parasitological and histopathological examination of these nodules. Without histopathological examination, cases of disseminated xanthogranulomatosis can easily be clinically misinterpreted as mycobacterial or fungal granuloma, considering that this form of inflammation presents with tumourous masses and extensive adhesions to adjacent organs [41]. Therefore, disseminated xanthogranulomatosis often remains undiagnosed and its prevalence in avian medicine is likely underestimated. Nevertheless, a combination of visceral xanthogranulomatous inflammatory reaction, especially caused by bacterial pathogens such as mycobacteria, should not be completely ruled out [50].

Interestingly, all of the six birds, including the four birds with xanthogranulomatous inflammation, had intranuclear inclusion bodies in the kidneys and tested positive for PsAdV-2. The isolation of an adenovirus in avian species does not necessarily indicate that it is the aetiological agent of a disease [51]. Nevertheless, an association between adenovirus infection and excessive accumulation of fat has been observed in chickens in the field and could subsequently be demonstrated under experimental condition [24], and–just recently—an experimental study found that fowl adenovirus-4 (FadV-4) is capable of inducing hepatic steatosis in Leghorn Chicken [52], which are usually not predisposed to fat accumulation. This breed is used to create highly productive egg-laying hybrids, rather than for extensive meat production. A link to an aberrant lipid metabolism has also been described in human medicine, where an association of obesity/higher body weight with the presence of neutralizing antibodies to adenovirus Ad36 was seen in several studies [53,54,55,56]. Particularly in children, Ad36 seems to be strongly associated with lipometabolic disorders [57,58]. In the present study, birds with disseminated xanthogranulomatous disease were all of young age (from one to three years). Therefore, PsAdV-2 infection may be considered as a trigger in the pathogenesis of disseminated xanthogranulomatosis in the observed cases.

Two other birds in our study tested positive for PsAdV-2 without xanthogranulomatous lesions. However, one of these individuals had yet another manifestation of a lipometabolic disorder, namely, hepatic steatosis. At four years of age, this individual was the oldest among the presented cases, while the second, 10-month-old bird without xanthogranulomatous alterations, was the youngest. The latter had, in addition to the siadenovirus infection, multiple concurrent pathologies. Adenoviral inclusion bodies in this bird were also seen in the liver and spleen, in addition to hepatocellular necrosis, which is in accordance with previous description of PsAdV-2 in Orange-bellied Parrots (*Neophema chrysogastor*) [12]. These two birds died within a few hours after hospitalisation at the clinic. In both cases, the respective companion animal died within a time span of two to five months, finally diagnosed with disseminated xanthogranulomatosis and PsAdV-2 infection. There are only a few reports on adenoviral shedding within aviaries/collections, which is supposed to represent between 30% and 77% of infected birds [12]. Considering the potential pathogenicity and the rapid transmission within an aviary, collections of Red-crowned Parakeets which tested positive for PsAdV-2 should be carefully monitored. Screening for adenoviral infections in birds is not yet undertaken routinely. In view of the persistent nature of at least some adenoviruses, spread of infection cannot be controlled by quarantine alone, thus rendering dissemination likely [14]. Siadenovirus-DNA can be detected from cloacal swabs and droppings, suggesting a fecal/urinary-oral transmission [12]. Furthermore, vertical transmission has been demonstrated for some adenovirus strains in chickens [59]. In the screening of live birds for adenoviral shedding, molecular biological analysis of droppings has been proven superior to testing of cloacal swabs [12], while in terms of post-mortem examination, the preferred organ to sample for consecutive PCR is the kidney. For histopathology, it should be kept in mind that inclusion bodies are not consistently detected [12]. The overall prevalence of siadenoviruses in psittacine birds is currently unknown, but the distribution of these viruses is likely to be much more extensive than previously recognized [14]. At least in the captive Orange-bellied Parrot population, it is reported to be endemic, with an overall suspected prevalence of 42% [12].

In Red-crowned Parakeets, so far, there is only one report of PsAdV-2 (subsequently referred to as psittacine siadenovirus F), stemming from a confiscated individual which tested positive in a pharyngeal swab sample [8]. Pathological lesions in association with PsAdV-2 in this species as yet have not been published, and to date, there are no reports on the prevalence of PsAdV-2 in collections of Red-crowned Parakeets. In other psittacine species, PsAdV-2 infections were reported as associated with chronic hepatopathy [15], acute death [13], lymphoplasmacytic pancreatitis, splenitis and hepatitis [18], weight loss and lethargy [16]. The majority of infections, however, were described as subclinical [12,14]. It seems that most strains of avian adenovirus are unlikely to produce diseases by themselves, and their role in disease development is more likely to be part of a multiple aetiology [51]. Factors that govern virulence have not been conclusively defined; nevertheless, a species-related pathology cannot be ruled out. Consequently, there may exist a high prevalence of PsAdV-2 infection in Red-crowned Parakeets, and findings of concomitant xanthogranulomatous inflammation might just be incidental. In view of the fact that not all human patients infected with Ad36 concurrently suffer from obesity or related disorders [53], it cannot be ruled out that PsAdV-2 infection in Red-crowned Parakeets in some cases induces a clinically manifest impairment of lipid metabolism. Considering that lipometabolic disorders are multifactorial diseases, it is difficult to identify to what extent PsAdV-2 infection contributes to their pathogenesis. Since PsAdV-2 do not have the E4orf1 gene associated with the induction of adipogenesis in cells, no comparative analysis can be performed. Further studies to understand the molecular pathomechanism are, therefore, necessary. Either both diseases are common in the Red-crowned parakeet, and as such often occur simultaneously (but without interaction), or—as suggested—PsAdV-2 is able to provoke lipometabolic disorders at least in the Red-crowned parakeet, resulting in disseminated xanthogranulomatosis. Further research is essential, birds with suspicion of granulomatous diseases should be carefully examined, including diagnostic imaging, haematology and blood chemistry; for final diagnosis, a histopathological examination of a biopsy sample from granulomatous masses would be necessary. If disseminated xanthogranulomatosis, or other lipometabolic disorders (such as hepatic lipidosis) are present, screening for PsAdV-2 should be initiated. In addition, screening for PsAdV-2 of clinical healthy individuals and flocks of Red-crowned Parakeets, including at least a thorough clinical examinations of the birds, would provide more insight into the prevalence, the virulence and ability of PsAdV-2 to induce lipometabolic disorders in Red-crowned Parakeets.

## 5. Conclusions

Disseminated xanthogranulomatosis seems to be common in Red-crowned Parakeets. Therefore, disseminated xanthogranulomatosis should be kept in mind as an important differential diagnosis to malign neoplastic processes, mycobacterial and fungal granulomas in Red-crowned Parakeets. Psittacine adenovirus 2 infections should be considered as a possible risk factor for acquiring disseminated xanthogranulomatosis at least in Red-crowned Parakeets. However, the causative significance of PsAdV-2 for disseminated xanthogranulomatosis in Red-crowned Parakeets remains to be determined. A limitation of the study is the small number of cases and the relatively unknown epidemiology of adenovirus in captive populations of Red-crowned Parakeets. In addition to the detection of adenoviral DNA in the infected tissues, other techniques such as immunofluorescence, immunohistochemistry or in situ hybridization would have been of great value to strengthen the possible association with the clinical diseases and their application should, therefore, be considered in future cases.

## Figures and Tables

**Figure 1 animals-12-02316-f001:**
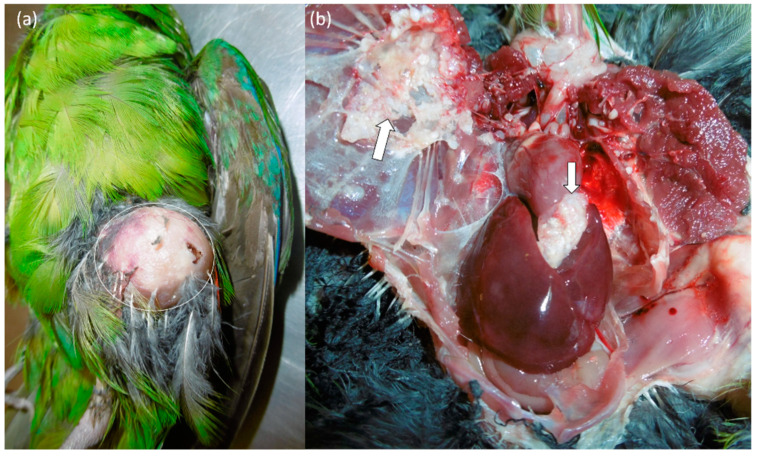
Red-crowned Parakeet (*Cyanoramphus novaezelandiae*) labelled as case no. 1. (**a**) Yellowish-colored, firm, fascial nodular mass, 1.6 cm in diameter, on the left leg (white circle). (**b**) Situs with multiple nodules between 0.1 cm up to 0.3 cm apical in the pericardium and serosa (white arrows).

**Figure 2 animals-12-02316-f002:**
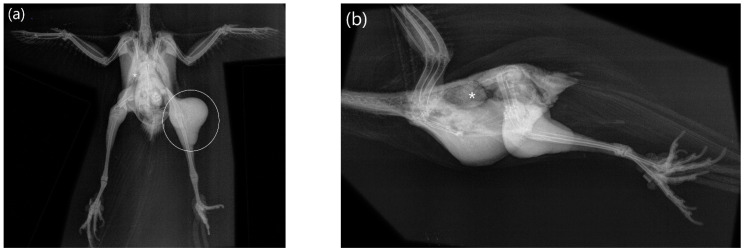
Total body radiographs of a Red-crowned Parakeet (*Cyanoramphus novaezelandiae*) labelled as case no. 1.: (**a**) Ventrodorsal projection with soft tissue shadowing of both knees, more pronounced on the left knee (white circle), with irregular decreased bone density and multiple osteolytic defects of the right distal femur and a round soft tissue shadowing within the right thoracic region, possibly associated with the right lung (white asterisk); (**b**) Latero-lateral projection with a soft tissue shadowing on the caudal edge of the lung (white asterisk).

**Figure 3 animals-12-02316-f003:**
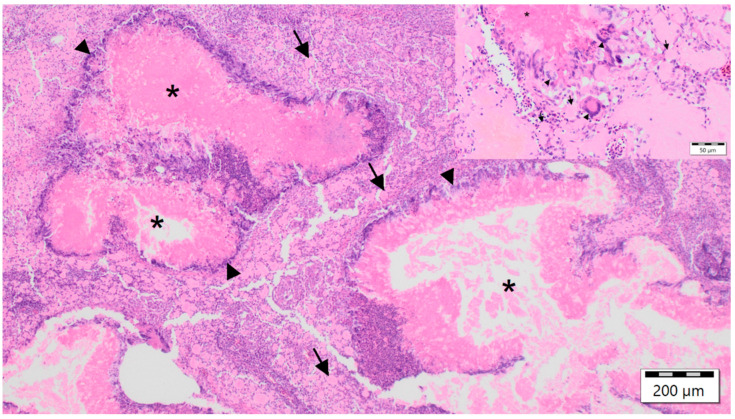
Micrograph of a subcutaneous fascial nodular mass from a Red-crowned Parakeet (*Cyanoramphus novaezelandiae*) labelled as case no. 1, characterized by diffuse foamy and cholesterol-laden macrophages (black arrows), multifocal fibrinous granulomas with central necrosis and fibrin (black asterisk), surrounded by multinucleated giant cells (black arrowheads), histiocytes, lymphocytes and heterophils. Haematoxylin and eosin staining, 100× magnification and inlay 400× magnification.

**Figure 4 animals-12-02316-f004:**
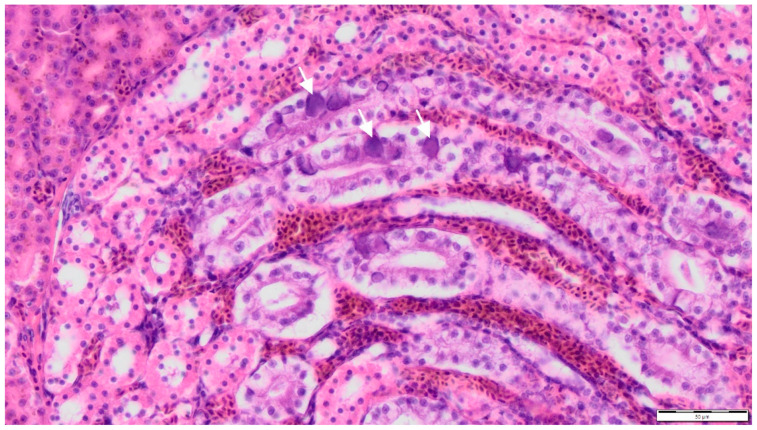
Micrograph of a kidney from a Red-crowned Parakeet (*Cyanoramphus novaezelandiae*) labelled as case no. 1 with large eosinophilic intranuclear inclusion bodies (white arrows) in the epithelial cells of the collecting ducts. Haematoxylin and eosin staining, 400× magnification.

**Figure 5 animals-12-02316-f005:**
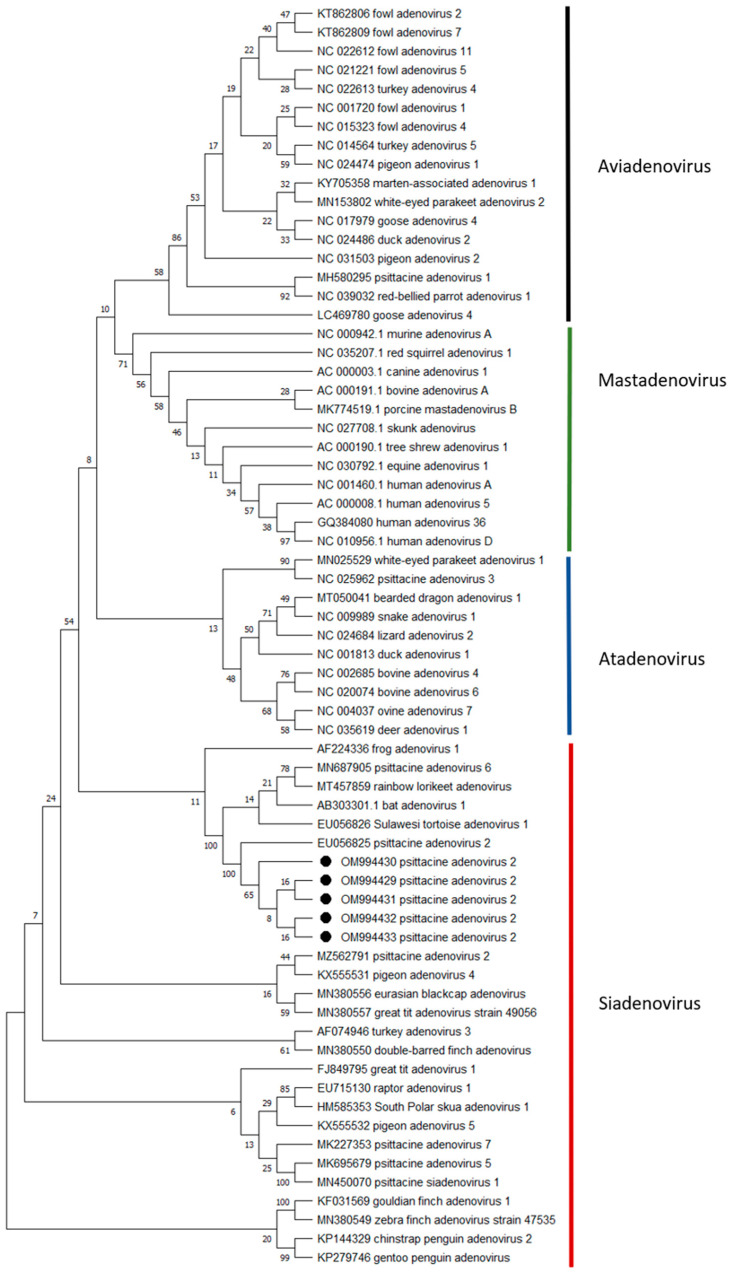
Phylogenetic analysis based on partial polymerase gene sequences (311 position, shorter sequences were excluded) of five sequences from this study in comparison to sequences of 62 reference sequences from NCBI. The five sequences belonging to this study are marked with a black circle.

**Table 1 animals-12-02316-t001:** Post-mortem findings of six Red-crowned Parakeets (*Cyanoramphus novaezelandiae*) with psittacine adenovirus 2 (PsAdV-2) infection, including age, weight, nutritional status, gender and additional findings. Locations of xanthogranulomas are listed in fascia (Fa), lung (Lu), liver (Li), spleen (Sp) and other organs such as air sacs, serosa, pericardium and mesenterium.

Case No.	Age in Months	Weight in g	Body Condition	Gender	Location of Xanthogranulomas					Additional Findings and Diagnoses
					Fa	Lu	Li	Sp	Ot	
1	24	73	3/5	Male	+	+	+	−	+	Overgrown beak and claws, feather loss, hepatomegaly, splenic and renal haemosiderin, *Procnemidocoptes janssensi*
2	36	72	3/5	Female	+	+	+	−	+	−
3 ^a^	10	58	2/5	Male	−	−	−	−	−	Overgrown beak and claws, hepatosplenomegaly, intravasal coagulopathy, acute necrotising splenohepatitis with amphophilic intranuclear inclusion bodies inside reticular cells and hepatocytes, lymphocytic depletion of the lymph follicles of the spleen, acute catarrhal duodenitis, subacute ulcerative proventriculitis and fibrinous fungal pneumonia
4 ^a^	12	79	3/5	Male	−	−	+	+	+	−
5 ^b^	48	69	4/5	Male	−	−	−	−	−	Overgrown beak and claws, steatosis hepatis
6 ^b^	30	49	1/5	Female	−	+	−	−	+	Subacute fibrinous conjunctivitis with isolation of *Pseudomonas aeruginosa*

^a^ Birds kept together in one household that died two months apart; ^b^ Birds kept together in one household that died five months apart.

## Data Availability

The datasets analyzed in the current study are available from the corresponding author upon reasonable request.

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
