# Peer review of "Case Series of Disseminated Xanthogranulomatosis in Red-crowned Parakeets (*Cyanoramphus novaezelandiae*) with Detection of Psittacine Adenovirus 2 (PsAdV-2)"

_animals, 2022, doi:10.3390/ani12182316_

Round 1

Reviewer 1 Report

Thank you very much for submitting the manuscript on a relatively unknown disease condition that effects psittacine species.  However, there are areas within the manuscript that require revision.

Line 87: Birds (animals) do not have "symptoms" only clinical signs.  Please correct throughout the manuscript.

Line 219: you go from Table 1 in the manuscript to Table 2 then back to a Table 1 "call out" in line 243.  The paragraph describing the haematological results should be after the Table 1 description.  In fact Table 2 is not needed and should be removed. You state the abnormal haematologic findings in the manuscript and the reader does not need to know the normal results.

Line 250: Nodules from which birds? Please state

Line 257: Again, nodules from which birds?

Line 265: It would be beneficial to remind the reader which cases had xanthogranulomas here.

Line 376: All birds? All six birds in the report? Please state the exact number here.

Line 379: Association between adenovirus and excessive accumulation of fat in chickens - you previously stated it was a chicken "model".  Were these chickens specifically selected because they were susceptible to generalized fat accumulation with OR without adenoviral infection? What breed of chicken was used and were they predisposed to fat accumulation?

Line 387: Please change "can" to "may". Therefore, PsAdV-2 infection "may" be considered.  Based on this report of 6 birds one cannot determine if the virus is a trigger or not for disseminated xanthogranulomatosis or not without more controlled studies.

Table 1: Under nutritional status it would be beneficial to state body condition score out of 5 or 9. "Reduced" does not give the reader any perspective of the birds body condition.  Also "nutritional status" is very vague and can have many meanings.  It appears you are describing body condition here so that should be the heading for this line in the table.

Table 2: This table is not needed as stated above. Please remove.

Figure 3: Identifiers (arrow, arrowheads) are needed in the image and inset image to show the reader what you are describing in the figure legend. A measuring bare is need in the image. Bar = micrometers

Author Response

Thank you for your very helpful and precise annotations and corrections.

Line 87: Birds (animals) do not have "symptoms" only clinical signs.  Please correct throughout the manuscript.

Thank you for that comment, it has been changed accordingly.

Line 219: you go from Table 1 in the manuscript to Table 2 then back to a Table 1 "call out" in line 243.  The paragraph describing the haematological results should be after the Table 1 description.  In fact Table 2 is not needed and should be removed. You state the abnormal haematologic findings in the manuscript and the reader does not need to know the normal results.

This has been adapted as suggested. We definitely agree, table 2 has been removed.

Line 250: Nodules from which birds?

The missing information has been added.

Line 257: Again, nodules from which birds?

The missing information has been added.

Line 265: It would be beneficial to remind the reader which cases had xanthogranulomas here.

The missing information has been added.

Line 376: All birds? All six birds in the report? Please state the exact number here.

Yes, all six birds had intranuclear inclusion bodies, the exact number has been stated as suggested.

Line 379: Association between adenovirus and excessive accumulation of fat in chickens - you previously stated it was a chicken "model".  Were these chickens specifically selected because they were susceptible to generalized fat accumulation with OR without adenoviral infection? What breed of chicken was used and were they predisposed to fat accumulation?

They were selected because an excessive fat accumulation in chickens infected with adenovirus has been observed in the field. They used Leghorn broiler in that study, which are usually not predisposed to fat accumulation. This breed is usually used to create highly-productive egg laying hybrids, rather than for extensive meat production.

Line 387: Please change "can" to "may". Therefore, PsAdV-2 infection "may" be considered.  Based on this report of 6 birds one cannot determine if the virus is a trigger or not for disseminated xanthogranulomatosis or not without more controlled studies.

We agree to that comment, it has been changed accordingly.

Table 1: Under nutritional status it would be beneficial to state body condition score out of 5 or 9. "Reduced" does not give the reader any perspective of the birds body condition.  Also "nutritional status" is very vague and can have many meanings.  It appears you are describing body condition here so that should be the heading for this line in the table.

Thank you for that comment, the body condition score has been added. 

Table 2: This table is not needed as stated above. Please remove.

As suggested, the table has been removed.

Figure 3: Identifiers (arrow, arrowheads) are needed in the image and inset image to show the reader what you are describing in the figure legend. A measuring bare is need in the image. Bar = micrometers

Thank you for the suggestions, the changes have been made accordingly.

Reviewer 2 Report

The study describes a case series of disseminated xanthogranulomatosis in red-crowned parakeet with a possible involvement of PsAdV-2 infection. PsAdV-2 is known to be highly prevalent in the psittacine birds worldwide. My only concern would be the relatively lower number of the cases. Besides, detection of the Adenoviral DNA in the affected tissues (IF/ISH/IHC) would be of great value to strengthen the possible association with the clinical diseases. Nevertheless, the study will be very useful for clinicians and scientists working on pet birds.

Author Response

Thank you very much for that precious considerations. It´s true, that up to date the sample size is relatively low, maybe in future with gaining knowledge and thorough diagnostic work up of suspected cases advanced numbers will be described. We totally agree that other techniques such as IF/ISH/IHC would provide more information; the methods will be established for future cases. We are grateful for that comment. 
